# Modelling and Mapping Urban Vulnerability Index against Potential Structural Fire-Related Risks: An Integrated GIS-MCDM Approach

Sepideh Noori [1], Alireza Mohammadi [1,*], Tiago Miguel Ferreira [2], Ata Ghaffari Gilandeh [1] and Seyed Jamal Mirahmadzadeh Ardabili [3]

[1] Department of Geography and Urban Planning, Faculty of Social Sciences, University of Mohaghegh Ardabili, Ardabil 56199-11367, Iran

[2] College of Arts, Technology and Environment, School of Engineering, University of the West of England (UWE Bristol), Frenchay Campus, Bristol BS16 1QY, UK; tiago.ferreira@uwe.ac.uk

[3] Department of Urban Planning, Islamic Azad University, Tabriz 51579-44533, Iran

* Correspondence: a.mohammadi@uma.ac.ir

**Abstract:** Identifying the regions with urban vulnerability to potential fire hazards is crucial for designing effective risk mitigation and fire prevention strategies. The present study aims to identify urban areas at risk of fire using 19 evaluation factors across economic, social, and built environment-infrastructure, and prior fire rates dimensions. The methods for "multi-criteria decision making" (MCDM) include the Analytic Hierarchy Process for determining the criteria's importance and weight of the criteria. To demonstrate the applicability of this approach, an urban vulnerability index map of Ardabil city in Iran was created using the Fuzzy-VIKOR approach in a Geographic Information System (GIS). According to the findings, about 9.37 km$^2$ (31%) of the city, involving roughly 179,000 people, presents a high or very high level of risk. Together with some neighbourhoods with low socioeconomic and environmental conditions, the city centre is the area where the level of risk is more significant. These findings are potentially very meaningful for decision-makers and authorities, providing information that can be used to support decision-making and the implementation of fire risk mitigation strategies in Ardabil city. The results of this research can be used to improve policy, allocate resources, and renew urban areas, including the reconstruction of old, worn-out, and low-income urban areas.

**Keywords:** risk analysis; structural fire risk; urban vulnerability index; mapping; GIS-MCDM; city

## 1. Introduction

Urban fire risk is the possibility of damage to people's life safety, property loss, and the threat to public security caused by the interaction between fire accidents and urban vulnerability, and the possibility of negative consequences or likely loss such as the breaking up of economic activities and environmental destruction [1]. Urban fire risk is a tremendous challenge to sustainable urban development, especially in low-income countries [2,3]. It causes damage to urban buildings and infrastructure and poses a major hazard to inhabitants' lives and property. The cost of fire damage is disproportionately high in major cities and highly inhabited areas [4]. According to the World Health Organization (WHO), around 3 million fires occur worldwide, with approximately 180,000 people dying each year [5]. Furthermore, most of these catastrophes occur in large cities in low-income countries [6,7] and many economic, social, and environmental conditions in these places raise the risk of possible fires [8]. While total fire prevention is virtually impossible, damages caused by potential building fires can be contained [9]. Modelling and predicting fire dangers is a crucial step in preventing fire damage in urban areas because it employs scientific frameworks to identify hazards [10] and assists local organisations in implementing appropriate

geographic data to cope with the damages associated with urban fires [11]. It is critical to identify the risk of fire in urban environments, particularly for rescue agencies in major cities [12]. Discovering and presenting high-risk regions for future fires would assist local organisations in taking effective measures to lower the risk, allocate resources in a more efficient manner, and distribute fire protection infrastructure throughout the city.

While resilience is the capacity of the system to withstand a major disruption within acceptable degradation parameters and to recover within an acceptable time, as well as composite costs and risks, vulnerability is the manifestation of the inherent states of the system that can be subjected to a natural or human-related hazard or be exploited to adversely affect that system. Contrarily, risk is based on probability and is determined by the likelihood and seriousness of unfavourable consequences [13]. In this essay, we take a closer look at fire risk and urban vulnerability to fire. Urban vulnerability is the result of the interaction of a number of disadvantages. Usually, the more vulnerable and distressed areas lack basic services and have a higher number of obsolete buildings, unfavourable social characteristics, vulnerable people, and more prominent social and environmental differences [14,15]. Urban vulnerability in this study refers to the possibility of fires in urban areas where there are poor environmental conditions, disrespect for building engineering requirements, and inadequate urban planning standards. Areas with poor socioeconomic conditions are included as well.

Fires are mostly spatial, meaning that they can be modelled and mapped [16], and that Geographic Information Systems (GIS) can be efficiently used to identify, manage, and anticipate fire events [17]. In fact, modelling and identifying high-risk and sensitive urban areas using spatial metrics [18] is an essential step towards reducing the probability of human and material losses resulting from fire events. Kernel Density Estimation (KDE), Monte Carlo Simulation models (MCS), geographically weighted regression (GWR), and other models of spatial analysis (Cluster analysis) have all been employed in recent years to identify vulnerable urban areas in terms of likely fire outbreaks in urban residential areas [19]. The framework for analysing the geographical patterns of probable fires is provided by identifying vulnerable urban areas and high-risk locations.

Over the last few years, a significant amount of research has been conducted on the spatial analysis and identification of factors impacting the increase of fire danger in urban areas. In the United States, the onset of research in this field dates back to the early 1980s [20,21] with studies primarily focusing on the influence of demographic and socioeconomic characteristics to assess fire risk. According to a Swedish study, the risk of probable fires rises dramatically in urban areas with larger building complexes, particularly when associated with other physical and social vulnerability factors, such as degraded buildings, overcrowded houses, and elderly inhabitants [22].

Some other investigations in Khulna, Bangladesh, found that, along with socioeconomic factors, built environment variables and urban infrastructure such as building quality, distance from high-voltage power plants, distance from fire stations and infrastructure, type of land use, and distance from warehouses and fuel storage or distribution centres all play a role in reducing or increasing the potential risk of fire in urban buildings [23]. According to a study conducted in Nanjing, China, fire risk is also more significant in downtown regions with a high concentration of commercial and economic activity [24]. Another study conducted in Helsinki, Finland, found that although the structure of fire risk distributions is highly variable, socioeconomic, and physical aspects of the urban neighbourhoods (such as building age and quality) have a direct influence on the increase or decrease of fire risk [25]. In addition, a study conducted in Nanjing, China, reported that the risk of fire increased considerably in buildings with mixed land use [26]. Furthermore, a study conducted in the Romanian city of Iași found that the city's outskirts are substantially more sensitive to fire than other metropolitan regions, with factors such as poor income, high population density, and inadequate physical structure all contributing to this urban vulnerability [27]. According to a study conducted in Zanjan, Iran, urban areas with a higher number of tall and old structures have a higher risk of fire [28]. Furthermore, economic-related as-

pects, such as low household income and high population density, have a critical and direct effect on raising the danger of fire in urban buildings, according to the research of 283 Chinese cities [29]. In a similar study conducted in southern Queensland, Australia, researchers found that in addition to economic and social factors, the distance or proximity to fire stations impacts the degree of sensitivity to potential fire threats in different urban regions [30]. Additionally, a study conducted in Seixal, Portugal, showed that the risk of fire is larger in the old downtowns, which are full of structures with mixed land activities and uses [31]. Deprivation, ethnicity, and sociocultural characteristics may all play a part in lowering or increasing the danger of future urban fires, according to a study conducted in the Midlands of the United Kingdom [32]. Similarly, poverty, population density, and poor building condition were all determined to be key variables in raising the danger of fire in a study conducted in Surabaya, Indonesia [33]. According to research conducted in Melbourne, Australia, while the city's fire risk has followed a complex pattern, the central section of the city has a higher risk of fire due to a mix of economic activities, land use, and property ownership [34]. Moreover, a study conducted in Melbourne, Australia, indicated that the risk of fire is higher in the suburbs with a higher population density and ethnic composition [35]. A similar study conducted in Melbourne validated the significance of high population density in increasing fire rates in urban areas [36]. In recent years, some studies have used GIS and MCDM methods to analyse the risk of residential and structural fires [37–39]. Table S1 in Supplementary File S2, in the supplement, lists and summarises the main findings of some relevant studies addressing fire risk in urban areas.

A review of early research suggests that most of the past research in this field has focused on space–time patterns or the link between variables that influence fire risk. Furthermore, the majority of research was conducted in developed countries or they employed fewer criteria to examine and estimate fire risk [17]. The texture and geometry of historic Middle Eastern cities, particularly in Iran, differ significantly from those of developed countries. However, no study of urban fire risk modelling and zoning utilising GIS approaches and a set of factors has been conducted in Iran. Identifying fire risk in urban areas and GIS-multi-criteria decision making (MCDM) analysis for fire risk mapping are instrumental in supporting informed decision-making and outlining efficient urban vulnerability mitigation strategies [40,41]. Efficient spatial deployment of urban fire stations and emergency services is highly desired to address the risk of modern urban fires [38]. Simple techniques are unable to predict fire risk in various geographic units due to the complexity of fire risk in urban environments. Then, the precise techniques for identifying high-risk locations must be applied. The integrated GIS-MCDA approach provides rapid, effective, and exclusive explanations to complex spatial complications [42]. In this regard, some researchers have employed methods based on the GIS-MCDM approach [43,44]. In this context, the main objective of this study is to apply an integrated GIS-MCDM approach to model and introduce high-risk urban regions in terms of fire occurrence in Ardabil city, located in the northeast of Iran. To model the vulnerability level in a GIS setting, 19 socioeconomic sub-criteria, built environment, facilities, and fire records were defined and applied.

## 2. Materials and Methods

### 2.1. Study Area

Over the period between 1990 and 2017, about 20,000 fires were reported in Iran's large and medium-sized cities. For example, in 2017, a fire destroyed a 17-storey Plasco commercial building in central Tehran, killing 25 people, wounding 235 others, and inflicting millions of dollars in damages [45]. Another very relevant Iranian town in terms of fire occurrences in Ardabil city. Between 2015 and 2020, an average of 300 structural fires were reported annually in the historical centre of this city [45]. Located in the northwest area of Iran, the city of Ardabil is the capital of the province with the same name. The town, used as a case study in this work, covers around 76 km$^2$ and is the house of about 530,000 inhabitants (about 7000 per km$^2$), according to the most recent census data [45].

Ardabil is divided into 5 administrative districts and 44 neighbourhoods. Regarding fire safety-related infrastructures, there are seven fire stations in the city, whose location is illustrated in Figure 1. Due to its physical and sociodemographic characteristics, fire combat is challenging in this city, particularly in the older parts of the city due to their spatial arrangement [46]. Figure 1 depicts the spatial density map (heatmap) of fire incidents (per hectare) as calculated using the KDE technique in QGIS a free and open-source GIS package [47].

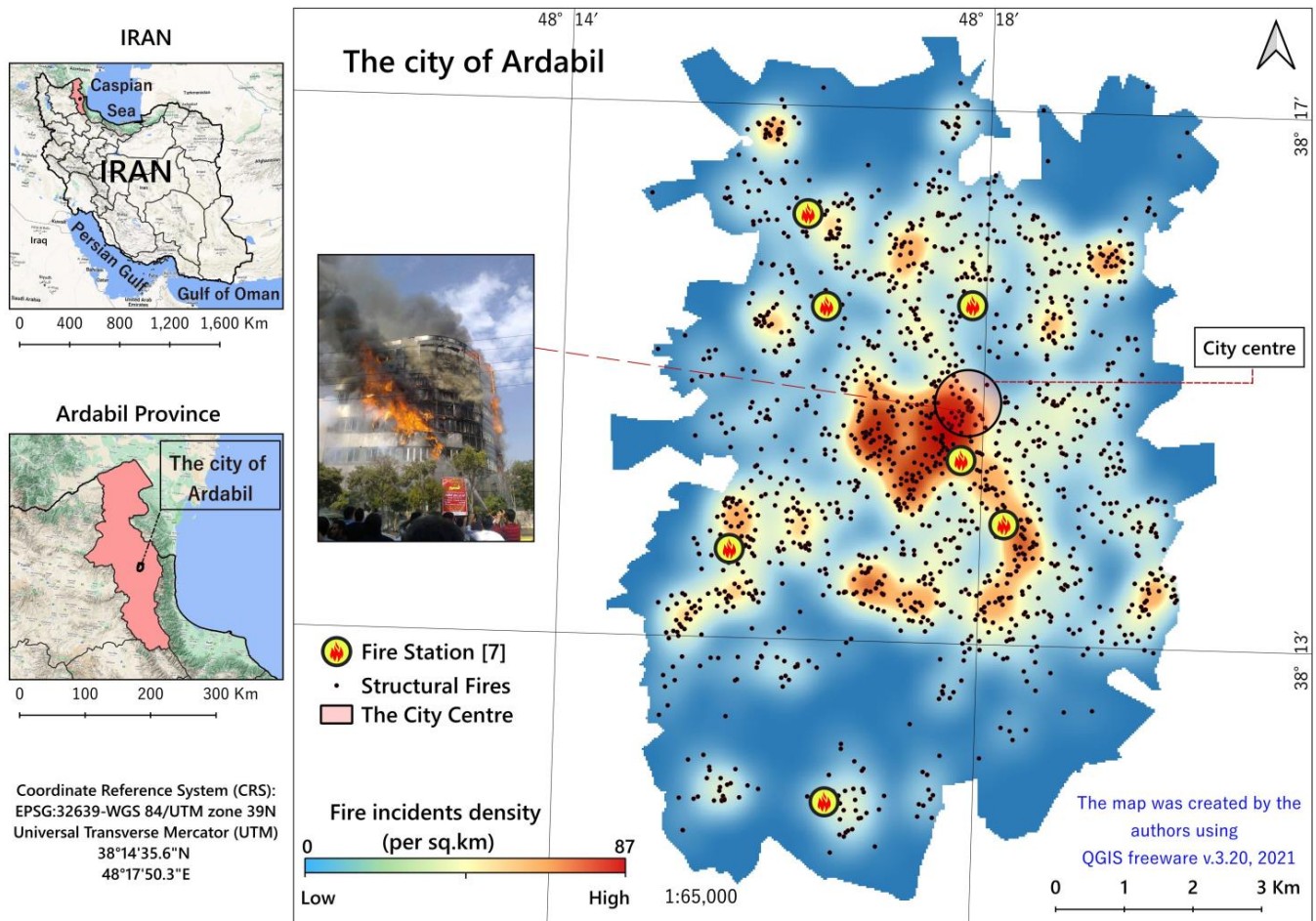

**Figure 1.** Map of Ardabil, Iran, showing the administrative divisions, fire stations, and the geographical distribution of fire incidents in the study area.

### 2.2. Data

Geodata sets and study criteria: From the combination of a thorough literature review (summarised in Table S1 of Supplementary File S2) and the objectives defined for the present study, 19 variables were isolated (outlined in Table S2 of Supplementary File S2) and used to evaluate the vulnerability of the buildings (about 250,000 building units) included in the study area. These variables are divided into four categories: (1) socioeconomic, (2) built environment, (3) infrastructure and urban facilities, and (4) previous fire incidence rates. The variables for each category are introduced in the following sections.

Socioeconomic: The Statistics Centre of Iran [45] provided raw data on socioeconomic factors such as population and household, number of elders, children, disabled people, and number of unemployed and illiterate people; see Table S2 in Supplementary File S2 and Figure 2 below.

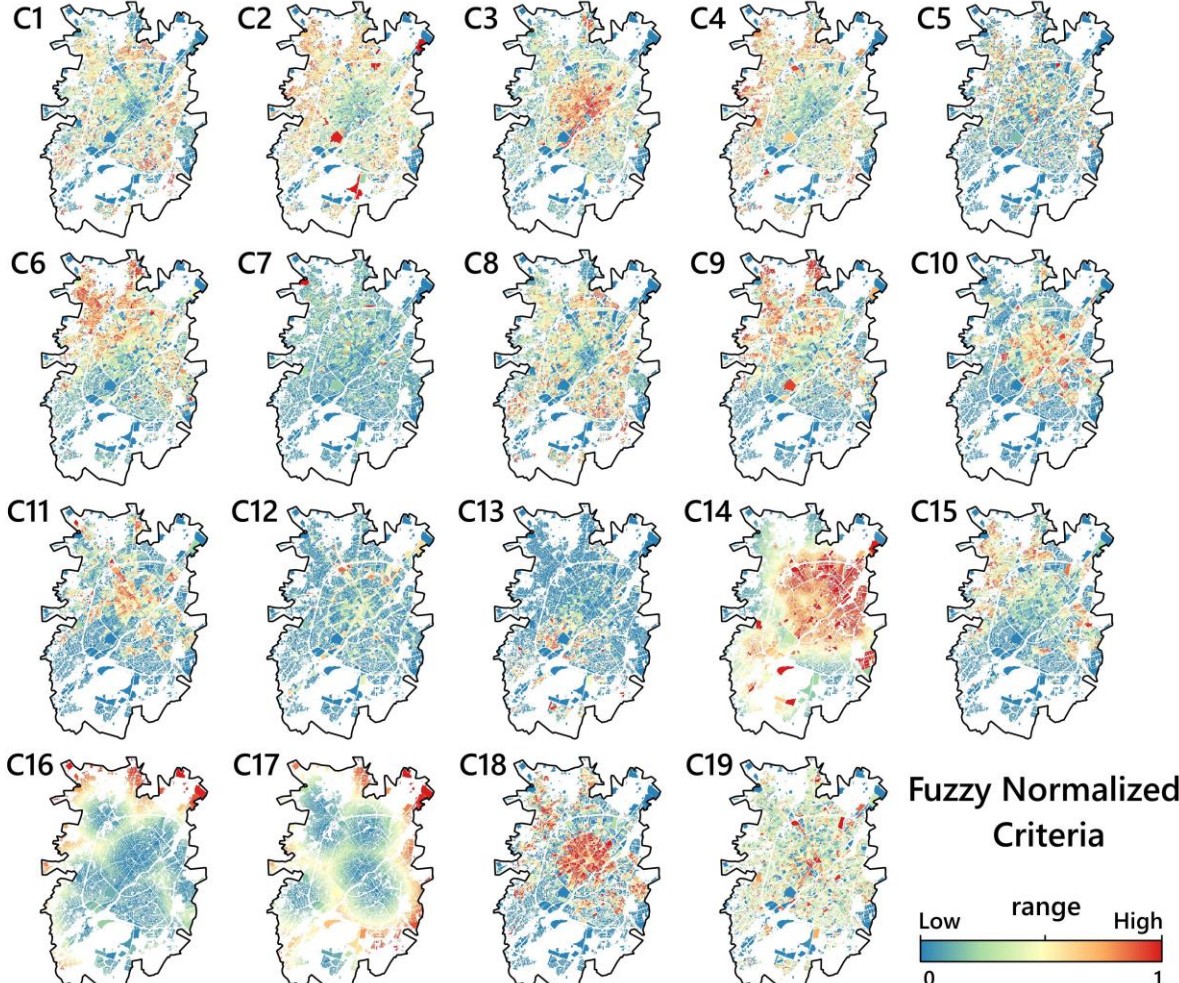

**Figure 2.** Spatial distribution map of fuzzy normalised criteria. C1: Population density, C2: Household dimension, C3: Old age ratio, C4: The ratio of the 14-year-old group and lower, C5: Disability ratio, C6: Illiteracy rate, C7: Unemployment rate, C8: Residential units' density, C9: The ratio of buildings made of non-durable materials, C10: The ratio of older buildings older than 30 years, C11: The ratio of worn-out and demolishing buildings, C12: Mixed land-use, C13: High-rise buildings ratio, C14: Buildings density with high fire incidence potential, C15: The ratio of small-sized property parts, C16: Euclidean distance from the hydrant valves, C17: Euclidean distance from fire stations, C18: The degree of permeability of the urban texture, C19: Previous fires rate.

Built environment: The type of materials used in the buildings (clay and mud, brick and iron), building quality (reparative, destructive, desolated), and number of storeys were taken from the statistical blocks of the most recent general housing census 2016 [45].

Infrastructures and facilities: Ardabil Municipality provided spatial data on the city road network (including blind roads and roadways less than 6 m wide) and landed segments, while Ardabil City Fire Department provided data on 46 fire hydrants and 7 fire stations in the city.

Location of fire incidents: Data on fire incidents from 2015–2020 were obtained from the files preserved at the Ardabil City Fire Department. The reason for choosing this period is twofold: the completeness, accuracy, and up-to-dateness of the data in this study area. Data that were missing, incomplete, or invalid were omitted from the analysis. Finally, 1488 fires in urban buildings (covering all urban land uses) were chosen as the final data and analysis' foundation. The Empirical Bayes Smoothing method was then used to obtain the rate of fire incidence per 100,000 people. Moreover, GeoDa v. 1.20.0.10 software [48] was used to determine the fire incidence Empirical Bayesian Smoothed (EBS) rates [49].

Within this study, several data from a variety of resources were taken into account. Based on the Iranian deviation systems, the urban blocks are considered as the smallest sector of the urban systems, which covers all information regarding residences, demography, and their specific characteristics. Because small geographical divisions such as urban blocks provide an accurate level for spatial analysis, they were chosen as the spatial basis of the analysis in the study area titled the highest resolution of urban geographical division. Point data were collected in a polygonal feature format layer in 6738 statistical blocks while establishing the UTM-Zone 39N coordinate system. The sub-criteria associated with each criterion utilised in this study (Supplementary File S2) were aggregated in the descriptive data table related to the polygon layer of statistical blocks using raw data and ArcGIS Desktop 10.8, and ArcGIS Pro 3.0.2 packages (ESRI, Redlands, CA, USA, 2022) were used to visualise our final model results [50].

*2.3. Methods, Tools, and Procedure*

2.3.1. Criteria Ranking and Weighting

In a GIS context, the maps for each criterion were created as raster maps. Because each index had a distinct size, the maps of each criterion were standardised using a fuzzy approach in a GIS environment to overcome this limitation, prepare the data, and execute MCDM methods. Different functions, such as S-shaped or J-shaped, as well as linear functions, are utilised in fuzzy standardisation. According to the nature and the linear relation between our criteria and the probability of fire assurances, the fuzzy S-shaped function was used to standardise benchmark maps in the present study (see Table 1). The fuzzy approach changes all raster layer's values and value rates to the same range of 0 (lowest index value) to 1 (highest index value), in Figure 2. The Fuzzy Overlay function in a GIS system was used to fuzzify and standardise the criteria for analysis [51]. After creating standardised fuzzy maps, the importance of each criterion was established using the numeric pairwise comparison approach by using Thomas L. Saaty's 1–9 Judgement Scale [52] and the opinions of ten experts. The final weight of the criteria was then calculated in the Expert Choice-11 software environment using the Analytic Hierarchy Process (AHP) method [53,54] (see Table 1). The compatibility ratio of the comparisons was calculated using Equations (1) and (2):

$$CR = \frac{CI}{RI} \qquad (1)$$

where CI represents the matrix compatibility vector, obtained from the following equation:

$$CI = \frac{\lambda max - n}{n - 1} \qquad (2)$$

where $\lambda_{max}$ is the largest matrix eigenvalue, RI is a randomness index for the matrix, and its value is proportional to the number of criteria in the matrix, with the number of criteria increasing the value. The pairwise comparison matrix's compatibility ratio should be smaller than 0.1. Otherwise, the preference judgments made are incoherent, and this incoherence should be addressed. Given that the RI ratio was equal to 0.09, the comparisons conducted to establish the importance of the criterion were confirmed. The weights obtained by the AHP method are given in the Table 1. According to experts, not all criteria are equally significant in predicting the likelihood of structural fire incidence, and some criteria, such as C9 (the ratio of buildings made of non-durable materials), C11 (the ratio of worn-out and demolished buildings), and C14 (building density with poor structural quality and a high risk of fire), have a higher importance.

**Table 1.** Summary statistics of criteria analysis.

| Criteria | | Statistics | | | | | |
|---|---|---|---|---|---|---|---|
| **Symbol** | **Criterion** | **Min** | **Max** | **Mean** | **SD** | **AHP Weights** | **Fuzzy Membership Function** |
| C1 | Population density | 0 | 86.1 | 2.03 | 2.22 | 0.032 | linear s-shaped |
| C2 | Household dimension | 0 | 10.7 | 0.24 | 0.22 | 0.022 | linear s-shaped |
| C3 | Old age ratio | 0 | 50 | 3.55 | 4.40 | 0.031 | linear s-shaped |
| C4 | The ratio of the 14-year-old group and lower | 0 | 50 | 11.81 | 8.90 | 0.035 | linear s-shaped |
| C5 | Disability ratio | 0 | 91.73 | 0.98 | 2.55 | 0.036 | linear s-shaped |
| C6 | Illiteracy rate | 0 | 91.46 | 8.23 | 8.49 | 0.02 | linear s-shaped |
| C7 | Unemployment rate | 0 | 0.34 | 1.48 | 2.25 | 0.019 | linear s-shaped |
| C8 | Residential units' density | 0 | 287 | 58.69 | 66.59 | 0.031 | linear s-shaped |
| C9 | The ratio of buildings made of non-durable materials | 0 | 100 | 10.71 | 19.79 | 0.069 | linear s-shaped |
| C10 | The ratio of older buildings older than 30 years | 0 | 100 | 13.80 | 24.12 | 0.051 | linear s-shaped |
| C11 | The ratio of worn-out and demolishing buildings | 0 | 100 | 19.74 | 26.85 | 0.099 | linear s-shaped |
| C12 | Mixed land-use | 0 | 0.71 | 0.04 | 0.10 | 0.055 | linear s-shaped |
| C13 | High-rise buildings ratio | 0 | 100 | 2.34 | 8.48 | 0.049 | linear s-shaped |
| C14 | Buildings density with high fire incidence potential | 0.84 | 4.88 | 2.88 | 0.46 | 0.114 | linear s-shaped |
| C15 | The ratio of small-sized property parts | 0 | 100 | 23.16 | 30.63 | 0.033 | linear s-shaped |
| C16 | Euclidean distance from the hydrant valves | 0 | 3907.24 | 949.05 | 660.7 | 0.066 | linear s-shaped |
| C17 | Euclidean distance from fire stations | 0 | 4091.57 | 1352.56 | 686 | 0.08 | linear s-shaped |
| C18 | The degree of permeability of the urban texture | 0 | 100 | 29.65 | 38.83 | 0.079 | linear s-shaped |
| C19 | Previous fires rate | 0 | 555 | 17.87 | 29.31 | 0.078 | linear s-shaped |

### 2.3.2. Fuzzy-VIKOR Method

VIKOR, as a prevailing MCDM method in the literature, ranks the alternatives based on the distance to the ideal condition [55]. Let $i \in \omega$ represent an alternative or raster cell in the set of alternatives ($\omega = \{1, 2, 3 \ldots m\}$) in which m is the last alternative. All cells in the study area are considered an alternative and based on cell value; they have the chance to be evaluated as risk cells for the projected locations. Given that $j$ is a criterion in the set of criteria in which $j$ is the last criterion, $x_{ij}$, then, is the preference value of alternative i in relation to criterion $j$. Let $f_{ij}$ be the normalised preference value of alternative $i$ in relation to criterion $j$, computable according to Equation (3):

$$f_{ij} = \frac{x_{ij}}{\sqrt{\sum_{i=1}^{m} x_{ij}^2}} \tag{3}$$

Using the $f_{ij}$ values, the maps with dissimilar scales and ideal solutions can be converted to the standard maps. The best $f_j^*$ value for the positive and negative criteria is calculated from the following Equation (4):

$$\begin{aligned} f_j^* &= {}^{Max}_{i} \, f_{ij} \text{ if it is a benefit-based function;} \\ f_j^* &= {}^{Min}_{i} \, f_{ij} \text{ if it is a cost-based function.} \end{aligned} \tag{4}$$

The worst $f_j^-$ value for the positive and negative criteria is calculated from Equation (5):

$$\begin{aligned} f_j^- &= {}^{Min}_{i} \, f_{ij} \text{ if it is a benefit-based function;} \\ f_j^- &= {}^{Max}_{i} \, f_{ij} \text{ if it is a cost-based function.} \end{aligned} \tag{5}$$

Let $S_i$ and $R_i$ indicate suitability and regret associated with alternative i, respectively. Then, related values are computable as Equation (6):

$$S_i = \sum_{i=1}^{n} w_i \frac{f_j^* - f_{ij}}{f_j^* - f_j^-}$$
$$R_i = Max\left\{ w_i \frac{f_j^* - f_{ij}}{f_j^* - f_j^-} \right\} \tag{6}$$

where, $w_i$ represents the weight of the *i*th criterion. The weight of each criterion was calculated through the Delphi method and was applied to each criterion in the GIS environment. The VIKOR value $Q_i$ that represents the maximum group benefits for alternative *i* can then be measured by Equation (7) for each alternative *i*:

$$Q_i = v\left[ \frac{S_i - S^-}{S^* - S^-} \right] + (1-v)\left[ \frac{R_i - R^-}{R^* - R^-} \right] \tag{7}$$

where

$$R^* = MaxR_i \,, R^- = MinR_i, \; S^* = MaxS_i \,, \; S^- = MinS_i$$

refers to the weight of criterion that ensures maximum group utility, and $(1 - v)$ refers to the weight of the minimum regret in dissent. The value of $v$ varies between 0 and 1; however, it is often taken as 0.5.

For an alternative to be preferable, its preference should be confirmed by the associated value of $Q_i$ addition to either of $S_i$, $R_i$, $Q_n$, with the smallest value expressed as the best option among alternatives. In this study, $Q_n$ is the location value of each alternative or cell in GIS. The least-valued alternative (point) is the most appropriate alternative to be selected. In the VIKOR method, if $A_1$ and $A_2$ are ranked first and second alternatives, respectively, to specify the value of "Q" (the chance that a fire may occur in each cell), Equation (8) should be satisfied [55]:

$$Q(A_2) - (A_1) \geq \frac{1}{n-1} \tag{8}$$

In this study, however, the cells have been categorised by the $Q_i$ score of each alternative or cell.

The VIKOR method's conclusions reveal the degree of risk that urban buildings face from a potential fire incidence. In the VIKOR method, the greatest value (high risk) in the output units (cells) is 0 and the lowest value (less risk) is 1. In the final step, we reversed the values for visualisations in the urban vulnerability index map.

The Natural Breaks (Jenks) classifying method approach was used to prioritise probable fire risk into five categories: lower (values = 0.74–1), low (values = 0.6–0.74), moderate (values = 0.48–0.6), high (values = 0.35–0.48), and higher (values = 0.086–0.35) degrees of urban vulnerability. In the Natural Breaks (Jenks) method, the variance within each class is minimised while the variance between classes is maximised [51]. With natural breaks classification, Natural Breaks (Jenks) classes are based on natural groupings inherent in the data. Class breaks are created in a way that best groups similar values together and maximises the differences between classes. The features are divided into classes whose boundaries are set where there are relatively big differences in the data values. Natural breaks are data-specific classifications and not useful for comparing multiple maps built from different underlying information [51,56].

Figure 3 illustrates the methodology utilised in the preparation of the urban vulnerability index map in terms of fire risk in the study area.

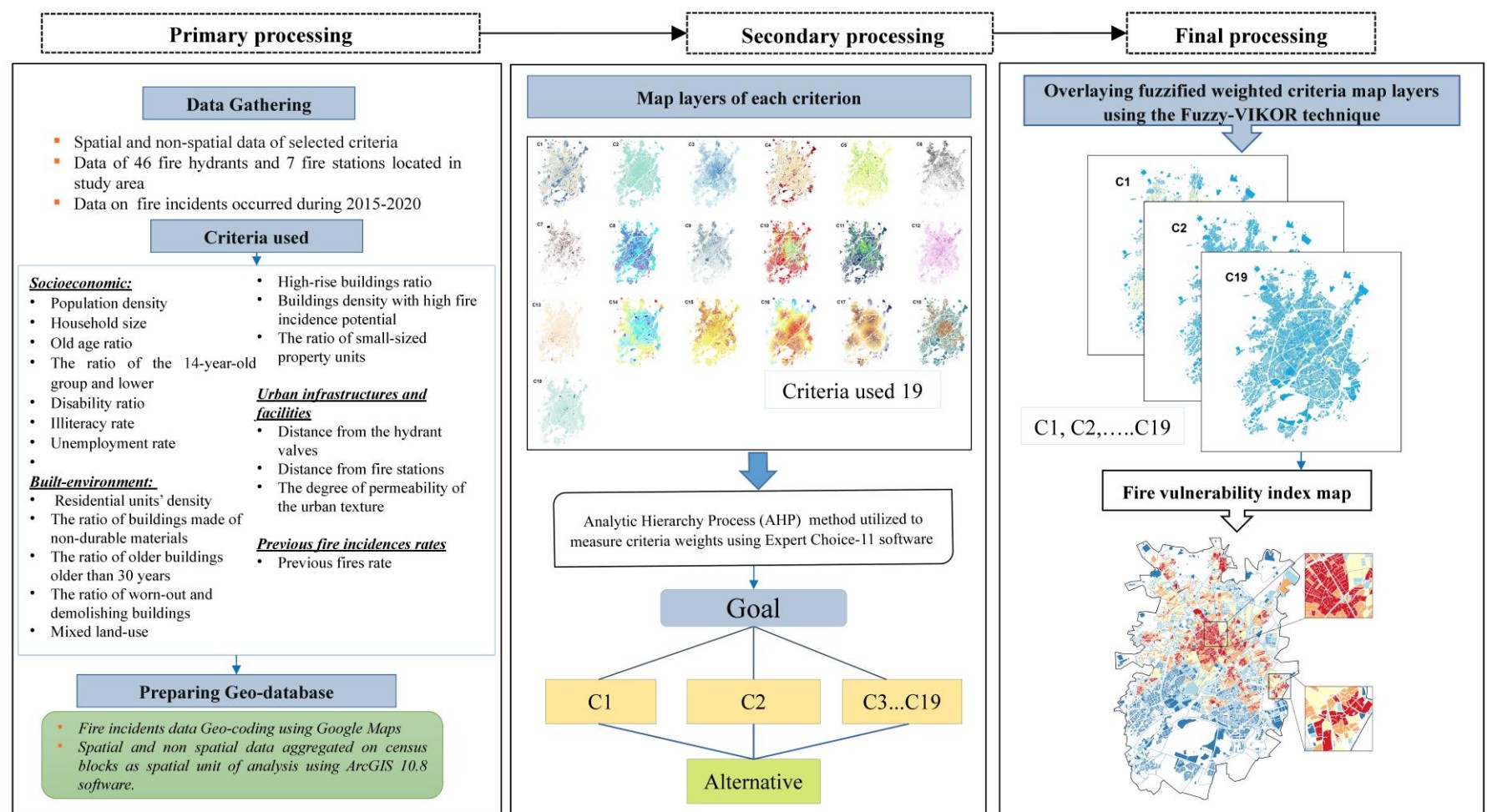

**Figure 3.** Flowchart illustrating the methodology used in various stages of the research for the preparation of the urban vulnerability index map in terms of fire risk in the study area.

2.3.3. Model Validation

To check the efficiency of the utilized GIS-MCDM model, the Spatial Linear Regression (SLR) approach [57] was used to measure correlation between the urban vulnerability index (result of our proposed model) and the spatial Kernel density of actual fire incidents in the study area using the TerrSet software V.19 (see: [57]). The following stage required mapping two variables using the Bivariate mapping method in ArcGIS Pro in order to visualise the model findings with real incident data. A bivariate map combines various sets of symbols and colours to represent two related but dissimilar variables on a map. It serves as a straightforward approach to depict, graphically and precisely, the link between the two spatially distributed variables. It is also simple to evaluate how two attributes change in respect to one another using this map [51].

**3. Results**

*3.1. Urban Vulnerability Index Map*

The aim of this study was to provide an urban vulnerability index map in terms of fire risk using integrated GIS-MCDM methods. The key result of this study is a vulnerability index map which is provided on the basis of our criteria used to map the risk of structural fire in the study area. Figure 4 depicts the vulnerability index map of buildings in different city areas within the urban blocks. Extracting the basic data from the VIKOR model's output map (Table 2) reveals that 639 blocks with a total size of 4.11 km$^2$ (13.62%) fall into the category of highly vulnerable locations, out of a total area of 30.18 km$^2$. These blocks are often found throughout the city's northern half, central district, and urban outskirts. A total of 930 blocks with a total size of 5.26 km$^2$ (17.43%) are classified as very vulnerable regions on this map. The blocks in this category are found throughout much of the city's northern half. As a result, 945 blocks with a total size of 5.31 km$^2$ (17.59%) are classified as moderately vulnerable. The blocks in this category are still found in most portions of the city's northern half (from east to west and from the centre to the north). The blocks in this category are found throughout much of the city's northern half. According to the findings, 1282 blocks with a total size of 5.97 km$^2$ (19.78%) fall into the low vulnerability index group. The blocks that fall into these two categories are frequently found at the city's outskirts, far from the downtown area. The blocks in this category are found throughout much of the city's northern half. Accordingly, very vulnerable locations include 2941 blocks with a total size of 9.53 km$^2$ (31.58%). The majority of these blocks are located on the city's southern side (from east to west and from the centre to south).

**Table 2.** Summary statistics of the urban vulnerability choropleth map given in Figure 4 based on Natural Breaks classifying method.

| Vulnerability Degree | Vulnerability Score | Number of Blocks | Area (sq.km) | Area (%) | Population | Population (%) |
|---|---|---|---|---|---|---|
| Higher | 0.086–0.35 | 639 | 4.11 | 13.62 | 72,471 | 13.83 |
| High | 0.35–0.48 | 930 | 5.26 | 17.43 | 106,716 | 20.37 |
| Moderate | 0.48–0.6 | 945 | 5.31 | 17.59 | 104,254 | 19.90 |
| Low | 0.6–0.74 | 1282 | 5.97 | 19.78 | 103,797 | 19.81 |
| Lower | 0.74–1 | 2941 | 9.53 | 31.58 | 136,663 | 26.09 |

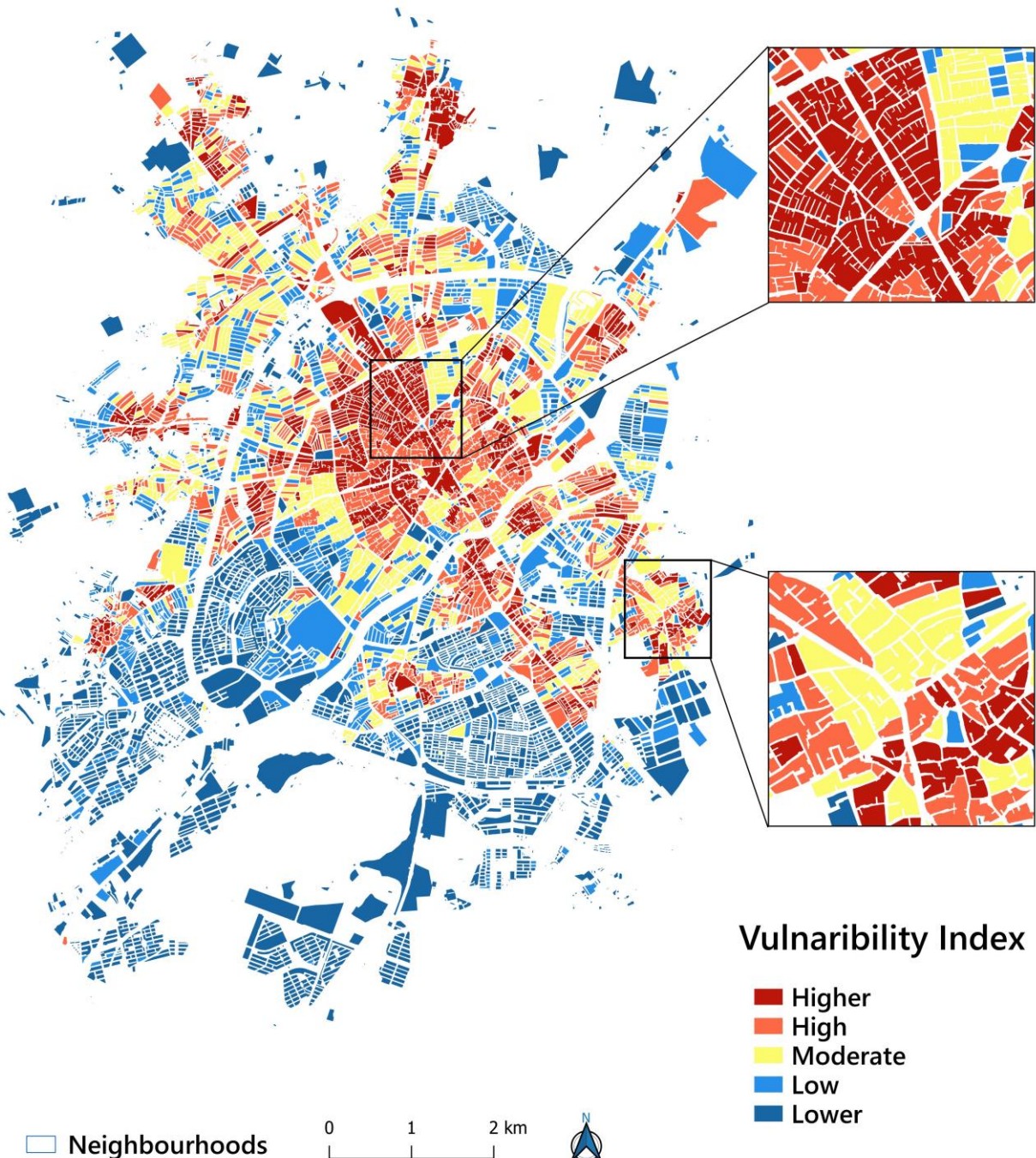

**Figure 4.** Spatial distribution map of areas with different degrees of urban vulnerability index against the potential structural fires-related risks.

### 3.2. Model Performance

We used the spatial linear regression correlation test value to compare the urban vulnerability index values with the spatial distribution of the actual fire incidence Kernel density (per hectare) in order to validate the model. The coefficient of determination ($r^2$) was equal to 100%, and the linear regression correlation's coefficient value was 1 (Figure 5). This significant outcome demonstrates that, using our criteria, the model used in this study was successful in determining the likelihood of fire risk over the sample period (2015–2020) and study area. Furthermore, the small variations between the VIKOR model's output map and the actual fire incident density (per hectare) spatial distribution (Figure 6) indicate a

good agreement between our model's measured urban vulnerability index values and the actual fire incident density (per hectare) spatial distribution in terms of the number of fire occurrences in the study area.

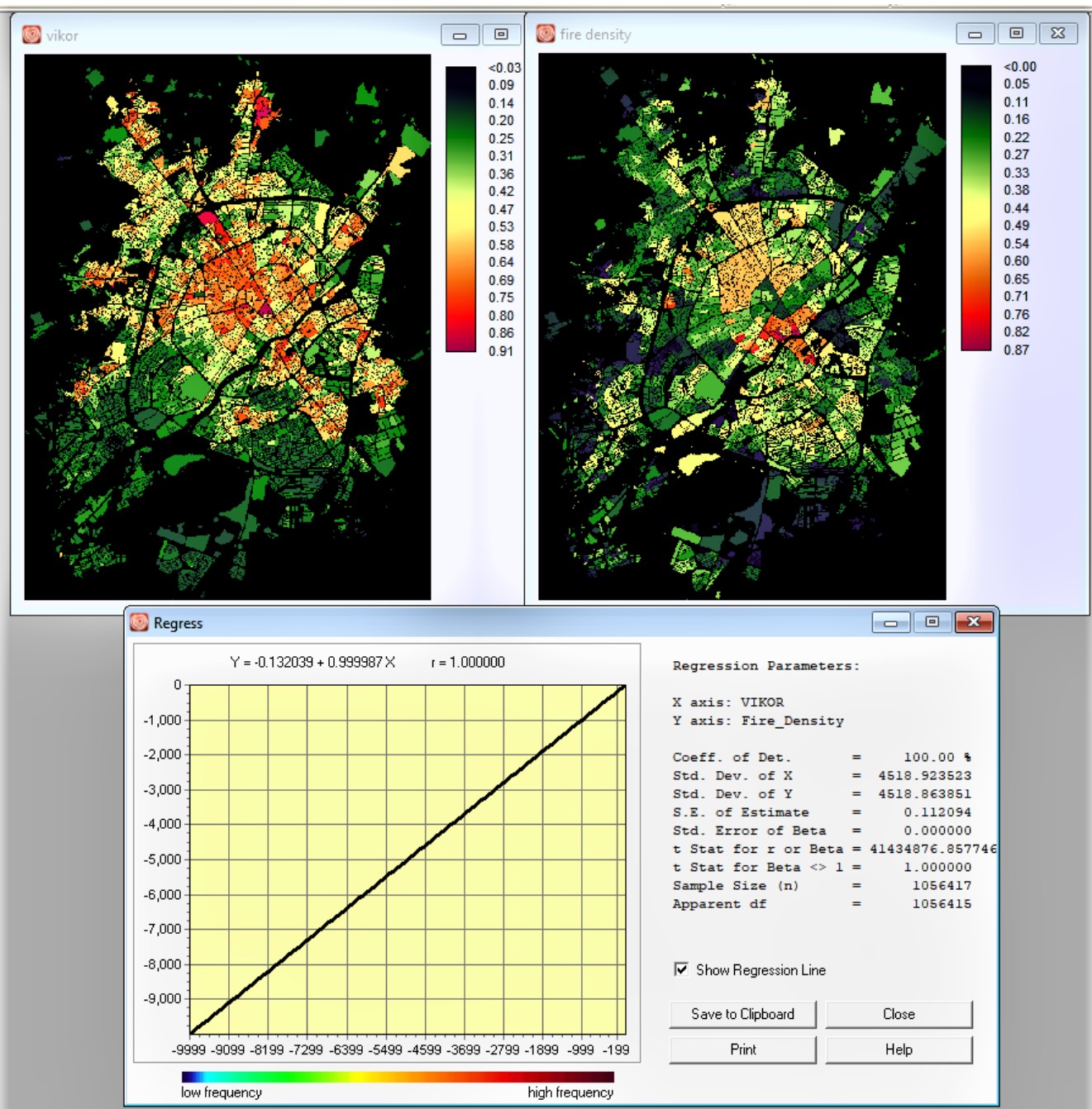

**Figure 5.** The spatial linear regression scatter plot of the spatial correlation between urban vulnerability against the potential structural fires-related risks index (*x*-axis) and actual fire incident per hectare values (*y*-axis). The plot generated in TerrSet (By: Clark Labs, Clark University, Worcester, MA, USA, 2022).

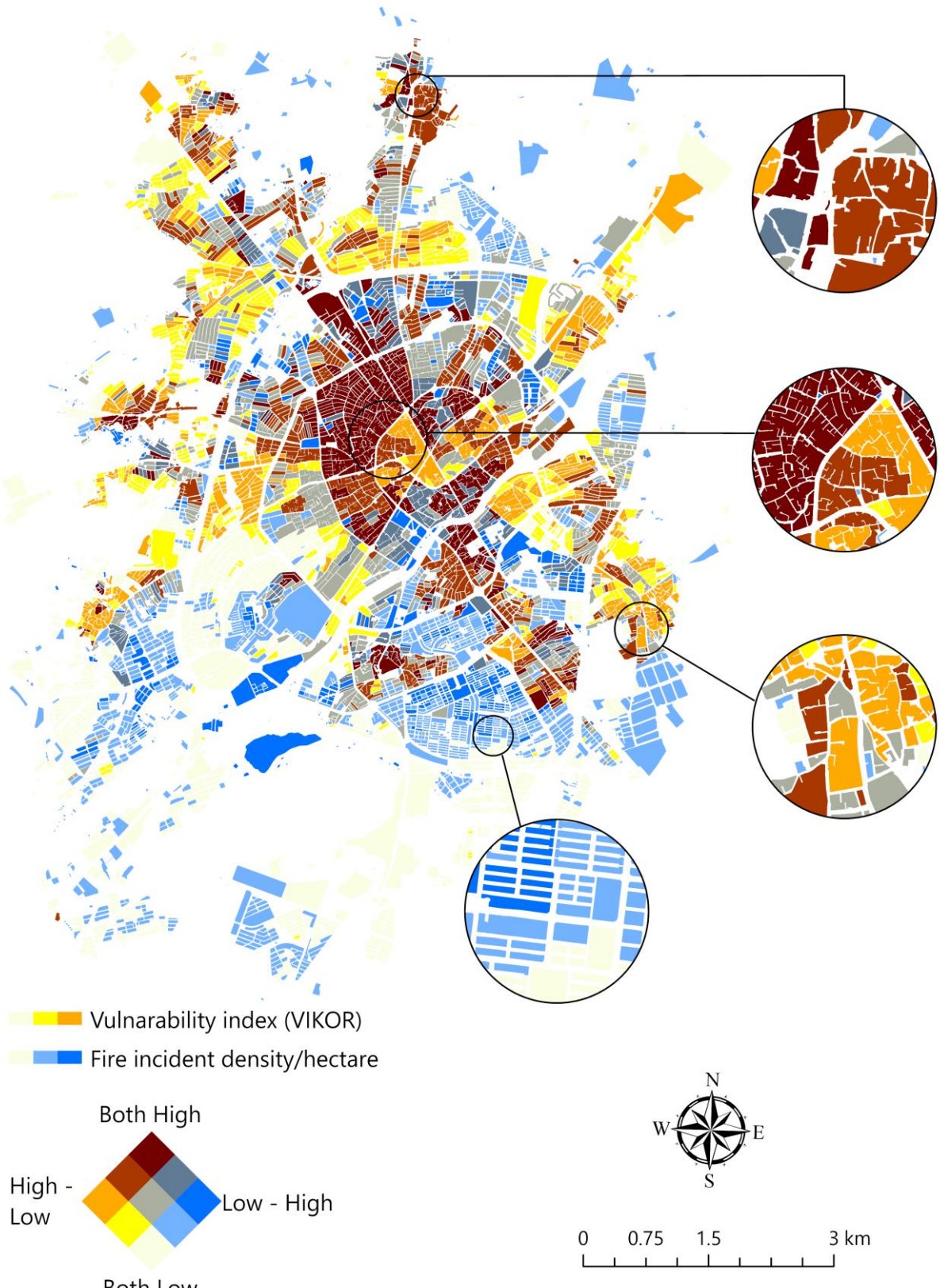

**Figure 6.** Spatial distribution bivariate map of urban vulnerability against the potential structural fires-related risks index (VIKOR model results) and fire incident per hectare original values. All the maps were generated in ArcGIS Pro 3.0.2 (ESRI, Redlands, CA, USA, 2023).

## 4. Discussion

The present study aims to model and prioritise urban regions in terms of their vulnerability to potential structural fires-related risks. The Integrated GIS-MCDM approach and 19 socioeconomic and built environment criteria were utilised to model and prioritise vulnerable urban regions. This study contains several important findings.

First, our GIS-MCDM Fuzzy-VIKOR model's output revealed that urban areas with high and extremely high vulnerabilities were increasingly spread from the city centre to the outskirts in the north, east, and west in terms of the spatial pattern (see Figure 4). This result is in line with the findings of Chhetri et al. [26] in southeast Queensland, who have shown that fires are distributed in a central–peripheral pattern. In addition, high-vulnerability areas in terms of potential fire risk frequently correspond to urban districts populated by low-income people (Figure 2 (C1)). The primary features of this portion of the city are the small-sized property (real state) units (Figure 2 (C15)) and high rates of unemployment (Figure 2 (C7)) and illiteracy (Figure 2 (C6)). This finding is in line with those obtained by Chhetri et al. [26,36], who observed that building fires are more common in older neighbourhoods and low-income residential areas on the outskirts of cities. Moreover, this finding is consistent with those of Zhang et al. in Nanjing, China. [3], since they observed a positive, strong, and significant correlation between poor income, unemployment, and illiteracy rates, as well as growing fire incidence rates in various urban areas. According to Rahmawati et al., [29] most poor settlements that are developed unintentionally with high population densities have a higher risk of fire incidence. Additionally, as Ardianto et al. [30] found, in urban areas, the overall socioeconomic and environmental variables have a significant effect in boosting fire incidence rates. In addition to socioeconomic situations, we found that most of the city's old and worn-out buildings are located in high-risk areas (Figure 2 (C11)), with non-durable materials utilised in their construction (Figure 2 (C9)).

Second, this study's results revealed that neighbourhood characteristics can be determinative in lowering or raising the risk of likely structural fire risk, and one component alone cannot play a role. In most vulnerable areas, the physical condition of buildings is relatively low (Figure 2 (C10, C11)), and the population in such places is ageing rapidly (Figure 2 (C3)). Previous studies have found that structural fires are becoming more common and pose a greater danger in urban areas with a high concentration of old and worn-out buildings [20,26,58,59].

Furthermore, we found that the city business district, known locally as the Bazar, is one of the focal points in the category of extremely vulnerable locations, as can be seen in the urban vulnerability index map presented in Figure 4. This place is at the city's most central location and serves a variety of purposes. Previous studies, such as Xia et al. [22], have found that mixed-used developments and commercial structures, particularly in central city areas, directly impacts the fire incidence rates.

Last but not least, the results displayed in Figure 4 make apparent the level of urban vulnerability in the city's outskirts, where spatial access to hydrant valves (Figure 2 (C16)) and fire stations (Figure 2 (C17)) is more reduced. In prior research conducted in Iran, Masoumi et al. [24] found that inadequate spatial access to urban amenities suited for firefighting might increase the probability of fire occurrence in densely populated regions.

Third, the integrated GIS-MCDM approach can be a useful tool for assessing urban vulnerability against probability fire-related risks in cities, as demonstrated in the model performance section (Figures 5 and 6). Because of its adaptability and capacity to interact with human inference and data-based processing, the combination of GIS-MCDM approaches allows for a more accurate prediction of fire risks, as our research revealed. The methods (such as the Point density or Kernel density methods) that ignore the human factor do not provide such a possibility. Moreover, by combining GIS with MCDM methods, it is feasible to combine a number of parameters and obtain more accurate results. In this study, other variables, such as the number of high-rise buildings, the proportion of children aged 14 and under, the population disability rate, the mixed land use coefficient, and the history of previous fires, were included in the integrated GIS-MCDM model as additional

variables. However, according to the study's findings based on the opinions of experts, their importance in the final output was lower; in other words, they were less active on a large scale in justifying and explaining the increase or decrease of urban vulnerability. Moreover, although some socioeconomic characteristics, such as population density and built environment variables, such as building age or impermeability and granularity of property components, have a significant influence in raising the risk of structural fire in the study area, as earlier studies [24,26,30] have pointed out, the set of a geographical area's circumstances and features can reduce or increase the risk of future structural fires.

### 4.1. Policy Implications

Firstly, it is proposed that the city council and municipality emphasise the renovation of ancient buildings, particularly in the downtown area, where the old Bazaar (a local name for the city business district) is located. Secondly, the buildings in the city's integrated villages and worn-out urban textures should be prioritised for rehabilitation. These are densely populated areas where low-income groups dwell, their buildings are of poor quality, and their quality of life is poor. Furthermore, it is critical to facilitate and improve physical access to fire stations for communities that are more vulnerable and have less access. In high-risk and extremely high-risk locations, new stations should be established. It appears that establishing dedicated routes for fire vehicles to approach and depart the city centre can help to lessen the damage caused by potential fires over time.

### 4.2. Limitations and Futures Research Strategy

The findings of this study have to be seen in the light of some limitations. We were unable to obtain information on household income and expenditures, as well as the status of building insurance. To circumvent this constraint, we examined additional social indices, including illiteracy and unemployment rates, as well as demographic data. The lack of access to urban banks and databases was another disadvantage of this study. Despite these limitations, we think this study has several strengths. The information for this study was gathered from a variety of sources by contacting various organisations. The use of a collection of socioeconomic data, the built environment, and characteristics linked to urban amenities in generating a map of urban vulnerability index to the risk of fire incidence were the study's strengths. We also prepared the final map using integrated MCDM-GIS hybrid approaches, which were less widely employed in earlier research for urban fires. Despite the limitations mentioned, the authors believe the methods utilised in this study will be valuable to academics and policy-makers working in the field of urban fire management.

## 5. Conclusions

The most vulnerable zones of the urban area analysed herein were identified by resorting to the hybrid MCDM-GIS approach and combining a number of characteristics that determine the fire risk in these areas. The findings revealed that using the hybrid MCDM-GIS approach to identify vulnerable zones in cities might be a useful tool. Urban vulnerability index maps concerning potential structural fires-related risks can assist in identifying elements that enhance fire risk and give useful insights into fire risk estimation and fire service management. Consequently, officials may utilise these maps to take preventative actions and allocate resources and infrastructure more effectively. Future research can look into the spatiotemporal patterns of urban fire phenomena, as well as more appropriate methods, such as spatial regression methods, to explore the relationship between different variables and fire rates and gain a better understanding of fire risk and the factors that influence it in urban areas. To simulate the risk of fire or urban vulnerability, future studies might employ more complex modelling approaches, such as agent-based modelling. We conclude that the methodological approach proposed in this study can be applied successfully to model and map fire risk in urban areas, with the potential to be applied in different urban contexts worldwide. In addition, the obtained results are of great importance to local stockholders, such as the municipality of Ardabil, authorities,

and decision-makers, in determining the spatiotemporal pattern of fire risk in the city and developing crisis risk programs.

**Supplementary Materials:** The following supporting information can be downloaded at: https://www.mdpi.com/article/10.3390/fire6030107/s1, Table: Supplementary File S1; Supplementary File S2. Refs. [60–68] are cited in Supplementary Materials.

**Author Contributions:** Conceptualisation, A.M. and A.G.G.; methodology, A.M. and S.N.; formal analysis, S.N.; geocoding and cleaning, S.N. and S.J.M.A.; writing, A.M. and S.N.; review, A.G.G., review and editing, T.M.F.; supervision, A.M. and T.M.F. All authors have read and agreed to the published version of the manuscript.

**Funding:** This research received no external funding.

**Institutional Review Board Statement:** Not applicable.

**Informed Consent Statement:** Not applicable.

**Data Availability Statement:** All the data used in this study to prepare the urban vulnerability index maps against the potential structural fires-related risks are available via Supplementary File S1.

**Acknowledgments:** We would like to acknowledge the Ardabil City Fire Department for sharing fire incidents data.

**Conflicts of Interest:** The authors declare no conflict of interest.

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
