# Peer review of "Modelling and Mapping Urban Vulnerability Index against Potential Structural Fire-Related Risks: An Integrated GIS-MCDM Approach"

_fire, doi:10.3390/fire6030107_

Round 1

Reviewer 1 Report

Urban fire risk is important to smart cities. The manuscript presents an integrated GIS-MCDM method for mapping urban fire vulnerability. The case study was used to verify the method, however, there are also several issued should be addressed.

1. The AHP was used to weight the 19 factors for mapping urban fire vulnerability. They used 10 experts, however, they should provide more details for the evaluations on the factors by each expert as it is important for AHP.

2. Factor C17, how to calcualte the distance? Roadnetwork distance or Euclidean distance?

3. How to get the fire incident density in 3.2?

4. The correlation between fire incident density and urban fire vulnerability is good to show their aggrement. The visual comparison between them is also required.

5. The related references should be cited. Such as, Chen, Y., Wu, G., Chen, Y., Xia, Z., 2023. Spatial location optimization of fire stations with traffic status and urban functional areas. Applied Spatial Analysis and Policy, 10.1007/s12061-023-09502-5.

Author Response

Dear Reviewer, 

We would like to express our gratitude for the time you dedicated to conducting a peer review of our manuscript. 

We have addressed each of the comments in detail in this letter and have made corresponding changes to the manuscript. We hope that these revisions meet your expectations. 

Reviewer # 1:

Comments and Suggestions for Authors

Urban fire risk is important to smart cities. The manuscript presents an integrated GIS-MCDM method for mapping urban fire vulnerability. The case study was used to verify the method, however, there are also several issued should be addressed.

  1. The AHP was used to weight the 19 factors for mapping urban fire vulnerability. They used 10 experts; however, they should provide more details for the evaluations on the factors by each expert as it is important for AHP.

Author response: Thanks for this comment. As you know, AHP is well known and has several steps. We combined all the comparisons of our experts and added final weights to Table 1. We clarified the wights by adding the AHP term to Table 1, column 7 in response to your suggestion. We also added a paragraph to section 3.21, following our methodology description. [Page: 6; Lines: 251-256 and Page: 7, Table1]

  1. Factor C17, how to calculate the distance? Road network distance or Euclidean distance?

Author response: Thanks for this comment. In this study and based on our purpose and spatial units of analysis, we measured distance using the Euclidean distance method in ArcGIS. For clarity, we have added the Euclidean type of measurement scale to table #1 after criteria C16 and C17. [Page: 7]

  1. How to get the fire incident density in 3.2?

Author response: Thank you for this comment. We used a Kernel Density Estimation method which spreads the known quantity of the population for each point out from the point of the location. Then it provides a more smoothed raster image as shown in Figure 1. In section 3.2, we added the word "Kernel" before density word in response to your insightful comment. [Page: 9; Line: 343]. We also mentioned the use of this technique in the "Study area" section. [Page 4; Line: 172].

  1. The correlation between fire incident density and urban fire vulnerability is good to show their agreement. The visual comparison between them is also required.

Author response: Thank you for this comment. Based on your kind request, we removed Moran’s I scatter plot and replaced it with an advanced and most appropriate method. We used Spatial Linear Regression to show the test association between two variables using the TerrSet software [Page: 9; Lines: 341-344]. Next, we also visualized the association of two variables in Figure 6. [Page: 13; Line: 402] and the revised text [Page: 12; Lines: 389-393].

  1. The related references should be cited. Such as, Chen, Y., Wu, G., Chen, Y., Xia, Z., 2023. Spatial location optimization of fire stations with traffic status and urban functional areas. Applied Spatial Analysis and Policy, 10.1007/s12061-023-09502-5.

Author response:  Thank you for your suggestion as well. We added this paper to our paper in the “Introduction” part, citation number 38. [Page: 3; Line: 144].

Reviewer 2 Report

The paper is written in good shape and is easy to understand. Make a minor revision; Fig.4 doesn't need an Index score as described in the table. Good luck. 

Author Response

Dear Reviewer,

We would like to express our gratitude for the time you dedicated to conducting a peer review of our manuscript.

We have addressed each of the comments in detail in this letter and have made corresponding changes to the manuscript. We hope that these revisions meet your expectations.

Reviewer #2

1) The paper is written in good shape and is easy to understand. Make a minor revision; Fig.4 doesn't need an Index score as described in the table. Good luck. 

Author response:  Thank you for your comment and your kindness. This suggestion was taken into consideration in the revised version and the corresponding figure has been modified. The term “Index score” has been removed in a new plotted figure (Figure 4). [Page: 12; Line: 384].

Reviewer 3 Report

Thank you for the opportunity to review the article "Modelling and mapping urban vulnerability index against potential structural fire-related risks: An integrated GIS-MCDM approach." I have completed reading the article and recommend the article for publication following very minor revision. Overall, I was very pleased with the quality and polish of the article -- it is a pleasure to review articles like this that require very little work to publish. The bivariate map is really stellar!

I have these recommendations to authors to improve the paper:

1. Multi-critieria decision making (MCDM) is abbreviated in the abstract, but not the paper. Geographic information science (GIS) is abbreviated in the paper, but not the abstract. I recommend giving the abbreviation in both the abstract and paper, but if word count is a concern, then I recommend consistency by providing the abbreviation in either the abstract or paper.

2. Figure 3 needs a little bit of work. I recommend using LibreOffice Draw to produce the figure without the distracting background colors and borders.

3. The most important area for elaboration is how the GIS-MCDM Fuzzy-VIKOR method might provide additional insight beyond, say, just examining building density. The GIS-MCDM method found that the bazaar (please check this spelling of this word on pg 409) was at the highest risk. In my limited understanding, this is to be expected. Essentially, how does the GIS-MCDM improve existing fire vulnerability indices? I recommend adding this to the literature review, as well as adding a few sentences to the discussion.

4. I recommend removing the border and background on Figure 1 (the main map, not the Iran or Ardabil maps) and remove the border on Figure 4 for consistency.

Author Response

Dear Reviewer,

We would like to express our gratitude for the time you dedicated to conducting a peer review of our manuscript.

We have addressed each of the comments in detail in this letter and have made corresponding changes to the manuscript. We hope that these revisions meet your expectations.

Reviewer #3

Comments and Suggestions for Authors

Thank you for the opportunity to review the article "Modelling and mapping urban vulnerability index against potential structural fire-related risks: An integrated GIS-MCDM approach." I have completed reading the article and recommend the article for publication following very minor revision. Overall, I was very pleased with the quality and polish of the article -- it is a pleasure to review articles like this that require very little work to publish. The bivariate map is really stellar!

I have these recommendations to authors to improve the paper:

  1. multi-criteria decision making (MCDM) is abbreviated in the abstract, but not the paper. Geographic information science (GIS) is abbreviated in the paper, but not the abstract. I recommend giving the abbreviation in both the abstract and paper, but if word count is a concern, then I recommend consistency by providing the abbreviation in either the abstract or paper.

Author response: Thanks for your valuable comment and kind opinion. We checked and checked all the abbreviations in the abstract and all the text and revised where it was needed. [Page: 1; Line: 21] and [Page: 2; Line: 141]  

  1. Figure 3 needs a little bit of work. I recommend using LibreOffice Draw to produce the figure without the distracting background colours and borders.

Author response: Thanks for this comment. Done. We used LibreOffice Draw to revise Figure 3. As you can see, we removed the background colours. [Page: 11; Line: 382].

  1. The most important area for elaboration is how the GIS-MCDM Fuzzy-VIKOR method might provide additional insight beyond, say, just examining building density. The GIS-MCDM method found that the bazaar (please check this spelling of this word on page 409) was at the highest risk. In my limited understanding, this is to be expected. Essentially, how does the GIS-MCDM improve existing fire vulnerability indices? I recommend adding this to the literature review, as well as adding a few sentences to the discussion.

Author response: Thanks for this valuable comment. Done. We checked and described the word “bazar” in the text [Page:16; Line: 483]. We also added a few sentences to the introduction [Page: 3; Line140-150] as well to the discussion parts [Pages:15-16; Lines: 461-467]

  1. I recommend removing the border and background on Figure 1 (the main map, not the Iran or Ardabil maps) and remove the border on Figure 4 for consistency.

Author response: Thanks for this comment. Done. We modified the figures [Page:4, Figure 1, and Page: 12, Figure 4, Page: 14, Figure 6] according to your recommendation.

Best regards, 

Authors,

Round 2

Reviewer 1 Report

All my concerns are addressed. It's ready for publication.

Author Response

Dear Reviewer,

Thank you very much for your kind words and news.

Best regards,

Authors, 
